# Distinct roles for innexin gap junctions and hemichannels in mechanosensation

Denise S Walker, William R Schafer*

MRC Laboratory of Molecular Biology, Cambridge Biomedical Campus, Cambridge, United Kingdom

**Abstract** Mechanosensation is central to a wide range of functions, including tactile and pain perception, hearing, proprioception, and control of blood pressure, but identifying the molecules underlying mechanotransduction has proved challenging. In *Caenorhabditis elegans*, the avoidance response to gentle body touch is mediated by six touch receptor neurons (TRNs), and is dependent on MEC-4, a DEG/ENaC channel. We show that hemichannels containing the innexin protein UNC-7 are also essential for gentle touch in the TRNs, as well as harsh touch in both the TRNs and the PVD nociceptors. UNC-7 and MEC-4 do not colocalize, suggesting that their roles in mechanosensory transduction are independent. Heterologous expression of *unc-7* in touch-insensitive chemosensory neurons confers ectopic touch sensitivity, indicating a specific role for UNC-7 hemichannels in mechanosensation. The *unc-7* touch defect can be rescued by the homologous mouse gene *Panx1* gene, thus, innexin/pannexin proteins may play broadly conserved roles in neuronal mechanotransduction.

## Introduction

Innexins are a family of proteins that form gap junctions in invertebrate neurons and muscle cells. Gap junctions allow free (though gated) movement of ions and small signaling molecules between the cytoplasm of the connected cells, resulting in electrical coupling and the propagation of signals such as $Ca^{2+}$ waves. Like in vertebrates, where gap junctions are formed from unrelated proteins called connexins, each invertebrate gap junction consists of two innexin hemichannels, each of which is a hexamer of constituent subunits (*Phelan and Starich, 2001*). The innexin families can be relatively large; for example, *C. elegans*, where innexins were originally identified, has 25 innexin genes. Different family members have distinct expression patterns, distinct gating properties, and differ in their ability to form homo- or hetero-hexamers and homo- or heterotypic gap junctions with specific partner hemichannels. There is thus enormous potential for variety, as well as asymmetry (rectification) in the relationships between partner cells (*Hall, 2019*; *Palacios-Prado et al., 2014*; *Phelan et al., 2008*).

In addition to their roles in gap junctions, the constituent hemichannels can also function independently as gated channels connecting the cell's cytoplasm with the exterior. Hemichannels have been shown to be gated by a variety of stimuli, including changes in extracellular pH, $Ca^{2+}$ concentration, or mechanical stimulation (*Hervé and Derangeon, 2013*; *Sáez et al., 2005*). Indeed, the vertebrate homologues of innexins, the pannexins, (*Baranova et al., 2004*; *Bruzzone et al., 2003*; *Yen and Saier, 2007*), are thought to function exclusively as channels (i.e. pannexons), not as gap junctions (*Sosinsky et al., 2011*). Humans have three pannexin genes, and it is becoming increasingly evident that they play an important role in a wide range of medically significant processes, such as apoptosis, inflammation, ischemia and tumour genesis (*Chiu et al., 2014*; *MacVicar and Thompson, 2010*; *Penuela et al., 2013*), as well as neuropathic pain (*Jeon and Youn, 2015*). Given the homology between innexins and pannexins, *C. elegans* represents an amenable system in which to gain a

*For correspondence: wschafer@mrc-lmb.cam.ac.uk

**Competing interests:** The authors declare that no competing interests exist.

greater understanding of these 'hemichannel' functions, as well as the role of gap junctions in the organisation and function of neuronal circuits.

Perhaps the best characterized innexin genes in *C. elegans* are *unc-7* and *unc-9*. Mutations in these genes were originally identified based on the uncoordinated locomotion phenotype caused by loss of UNC-7 or UNC-9 function (*Brenner, 1974*). Both genes are expressed in ventral cord motor-neurons and the premotor interneurons that promote forward or backward crawling (*Altun et al., 2009*; *Starich et al., 2009*). Heterotypic gap junctions between these premotor interneurons and motorneurons are important for controlling the balance between forward and backward locomotion as well as promoting coordinated sinusoidal locomotion (*Kawano et al., 2011*; *Starich et al., 2009*). They also play a central role in the regulation of sleep (*Huang et al., 2018*). In addition, UNC-7 has been shown to function as a hemichannel in motorneurons to promote neuromuscular activity through regulation of presynaptic excitability (*Bouhours et al., 2011*). UNC-7 has also been shown to function in the sensory circuit involved in nose touch, most likely through gap junctions in a hub-and-spoke electrical circuit (*Chatzigeorgiou and Schafer, 2011*). Both UNC-7 and UNC-9 are expressed in many additional neurons, where their functions have not been investigated.

Among the cells that express *unc-7* and *unc-9* (*Cao et al., 2017*; *Starich et al., 2009*; *Altun et al., 2009*; *Bhattacharya et al., 2019*) are the sensory neurons mediating gentle and harsh body touch. Six neurons (referred to as TRNs or gentle touch neurons) are involved in sensing gentle touch: the ventral AVM and PVM, and lateral pairs of ALMs and PLMs. A mechanosensory complex including the DEG/ENaC channel subunits MEC-4 and MEC-10 is required for gentle touch responses in all these neurons (*Bianchi, 2007*; *Bounoutas and Chalfie, 2007*; *Schafer, 2015*). The anterior touch neurons form a putative gap junction-coupled electrical network, with ALML and ALMR coupled to AVM, as well as to the locomotion circuit via AVDR (*Chen et al., 2006*; *Hall and Russell, 1991*; *Varshney et al., 2011*; *White et al., 1986*). In contrast, the posterior TRNs are not gap junction coupled, though they do make gap junctions with other neurons. The PVD neurons, which sense harsh body touch, also express both *unc-7* and *unc-9* but the extent to which they form gap junctions is unclear. While gap junctions were not detected previously (*Varshney et al., 2011*; *White et al., 1986*), this may be due to their complex, branched morphology; more recent analysis (*Cook et al., 2019*) identified a few gap junctions with motorneurons. Since pannexin one has been demonstrated to function as mechanosensitive, ATP releasing channels in multiple cellular contexts (*Bao et al., 2004*; *Beckel et al., 2014*; *Furlow et al., 2015*; *Richter et al., 2014*), this might suggest a role for innexin hemichannels in mechanotransduction in the PVDs and the TRNs.

In this study, we characterise the roles of two innexin subunits, UNC-7 and UNC-9, in *C. elegans* touch neurons. Both UNC-7 and UNC-9 are required for gap junction communication between the anterior TRNs, creating an electrically-coupled network that ensures a robust response to stimuli applied to either side of the animal. In addition, UNC-7 hemichannels play an essential role in gentle touch mechanosensation in both the anterior and posterior TRNs as well as harsh touch sensation in the PVD polymodal nociceptors. Heterologous expression of UNC-7a hemichannels in mechanically insensitive amphid neurons ASK or ASJ confers the ability to respond to nose touch, indicating that UNC-7a is sufficient as well as necessary to generate a mechanosensor. Since mouse pannexin one can functionally complement an *unc-7* null mutation, our results may suggest conserved functions of pannexins and innexins in other mechanosensory tissues.

## Results

### Innexins are required for mechanosensation and electrical coupling of touch neurons

At least three innexin genes – *inx-7*, *unc-7* and *unc-9* – have been shown to be expressed in the TRNs (*Altun et al., 2009*; *Starich et al., 2009*). We therefore used RNAi to investigate the role of these genes in mechanosensory activity. Since global knockdown could potentially have complex consequences, we used RNAi constructs expressed under the *Pmec-7* promoter, which drives strong touch neuron expression, to inactivate each innexin gene and imaged neuronal touch responses using a genetically encoded $Ca^{2+}$ indicator (*Kerr et al., 2000*; *Suzuki et al., 2003*). We observed that while knockdown of *inx-7* had no effect, knockdown of *unc-9* significantly reduced, and

knockdown of *unc-7* almost completely abolished, ALM touch responses (*Figure 1B–D*). Thus, *unc-7* and *unc-9* both appear to play roles in the gentle touch response exhibited by the TRNs.

The anterior touch receptor neurons are electrically-coupled through ALMR-AVM and ALML-AVM gap junctions; thus, these gap junctions could potentially influence touch responses. To investigate the importance of these electrical synapses in touch neuron activity, we first examined the consequences of laser ablating AVM, which would disrupt gap junction communication between ALML and ALMR. In wild type unablated animals, ALM responded robustly to ipsilateral (i.e. the left side for ALML) or contralateral (i.e. the right side for ALML) stimuli (*Figure 2A–C*). However, when AVM was ablated, ALM responded robustly only to ipsilateral stimuli. This indicates that electrical coupling between the anterior TRNs, via AVM, is required for touch neurons to respond to contralateral stimuli.

To assess the possible roles of *unc-7* and *unc-9* in this electrical coupling, we re-examined the effects of innexin RNAi, distinguishing between ALM responses to ipsilateral and contralateral stimuli (*Figure 2A–C*). We observed that knockdown of *unc-9* almost entirely abolished responses to contralateral stimuli but had little effect on responses to ipsilateral stimuli, suggesting that UNC-9 is an important constituent of the gap junctions coupling the ALMs through AVM. Consistent with this possibility, the combined effect of *unc-9* RNAi and AVM ablation was not significantly different from either *unc-9* RNAi alone or AVM ablation alone in either ipsilateral or contralateral neurons (*Figure 2A–C*). A loss-of-function mutant in *unc-9* showed a similar phenotype, with normal responses in ipsilateral neurons but reduced responses in contralateral neurons (*Figure 2*). Thus, disrupting gap junction communication appears functionally analogous to disrupting *unc-9*, supporting the hypothesis that *unc-9* plays an essential role in gap junction communication between the TRNs, and that this is perhaps its sole function in the TRNs. In contrast, *unc-7* RNAi significantly disrupted the responses in ALM to both ipsilateral and contralateral stimuli, suggesting that UNC-7 is required for mechanosensation per se, rather than simply contributing to gap junctions. As *Figure 2—figure supplement 1* shows, *unc-7* RNAi also severely disrupts responses in AVM, even when the stimulus was applied close to the AVM dendrite. Together, these results indicate that *unc-7* is required for robust mechanosensory responses in all three anterior TRNs.

## The mechanosensory function of UNC-7 is gap junction-independent

In principle, the mechanosensory defects seen in the anterior touch receptor neurons could result from changes in excitability due to a lack of UNC-7-containing gap junctions; alternatively, UNC-7-containing hemichannels could have a distinct, gap-junction-independent role in mechanosensation. To address these possibilities, we first investigated touch responses in the posterior TRNs PLML and PLMR, which are not connected by gap junctions, either directly or indirectly via PVM. In wild type animals, PLM responses to ipsilateral stimuli were extremely robust (nearly 100% responding), while less than half of contralateral stimuli generated responses (*Figure 2—figure supplement 2*), suggesting that the coordinated responses of the anterior TRNs are indeed gap-junction-dependent. When we measured responses in *unc-7* RNAi animals, we observed defective responses to both ipsilateral and contralateral stimuli, as seen previously for the anterior TRNs (*Figure 2—figure supplement 2*). Thus, *unc-7* appears to be required for the response in the posterior TRNs, despite their lack of gap junction interconnectivity.

Formation of gap junctions by innexins has been shown (*Bouhours et al., 2011*) to require four cysteines at the inter-hemichannel interface of UNC-7. When these residues are mutated, the innexin protein's ability to form functional gap junctions is disrupted, but its hemichannel function is intact. We therefore examined whether a 'cysless' mutant allele of *unc-7* could rescue the *unc-7* mechanosensory defect in touch neurons. As *Figure 3A–C* shows, ALM gentle touch responses are severely disrupted in *unc-7(e5)* animals, as we observed previously for *unc-7* RNAi. Expression of a wild type *unc-7* cDNA (isoform a, also known as UNC-7L *Starich et al., 2009*) under the control of the *mec-4* promoter significantly rescued this defect. In contrast to *Pmec-7*, which we used for RNAi and is expressed in various neurons in addition to the TRNs (*Mitani et al., 1993*), *Pmec-4* expression is exclusively TRN-specific. Thus *unc-7* is required cell-autonomously in the TRNs for its role in the response to gentle touch. Expression of the cysless mutant *unc-7a* cDNA very successfully rescued the *unc-7(e5)* gentle touch response defect, indicating that the TRN defect is related to hemichannel rather than gap junction activity. Expression of mouse *Panx1* (encoding *Pannexin 1*), but not *Panx2*, in the TRNs also successfully rescued the *unc-7* mechanosensory defect (*Figure 3A–C*), indicating

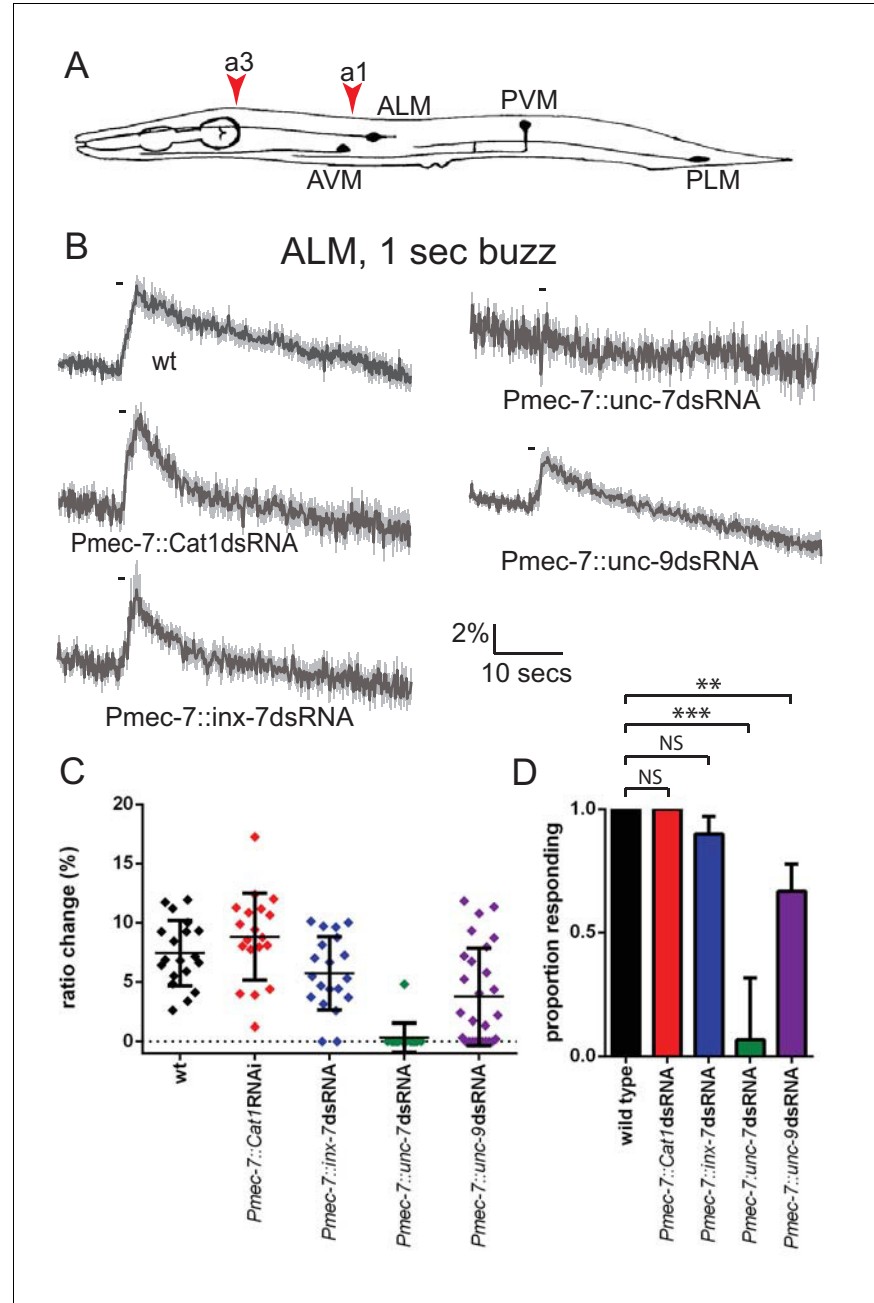

**Figure 1.** *unc-7* and *unc-9* function in touch responses in ALM. (**A**) Schematic showing positions of cell bodies and processes of the *C. elegans* touch receptor neurons. ALM and PLM are lateral pairs (left and right), of which only one of each is shown. Red arrowheads show stimulation sites. Except where stated, animals were stimulated at a3. As in later figures, we present average traces of % ratio change, a scatter plot showing individual ratio changes and a graph showing proportion exhibiting a $Ca^{2+}$ response. (**B,C,D**) Gentle touch responses recorded in ALM for wild type animals and animals expressing dsRNA under control of *Pmec-7*. (**B**) Average traces of % ratio change. Gray indicates SEM. (**C**) Scatter plot showing individual ratio changes (diamonds). Bars indicate mean ± SEM. (**D**) Graph showing proportion exhibiting a $Ca^{2+}$ response. Error bars indicate SE. *unc-7* (<0.0001) and *unc-9* (p=0.0062) RNAi are significantly different from wild type, while *E. coli* Cat1 (p=0.10.0) and *inx-7* (p=0.4872) RNAi are not, Fisher's exact test (N = 19, 19, 20, 15, 27, in the order shown in the graphs).

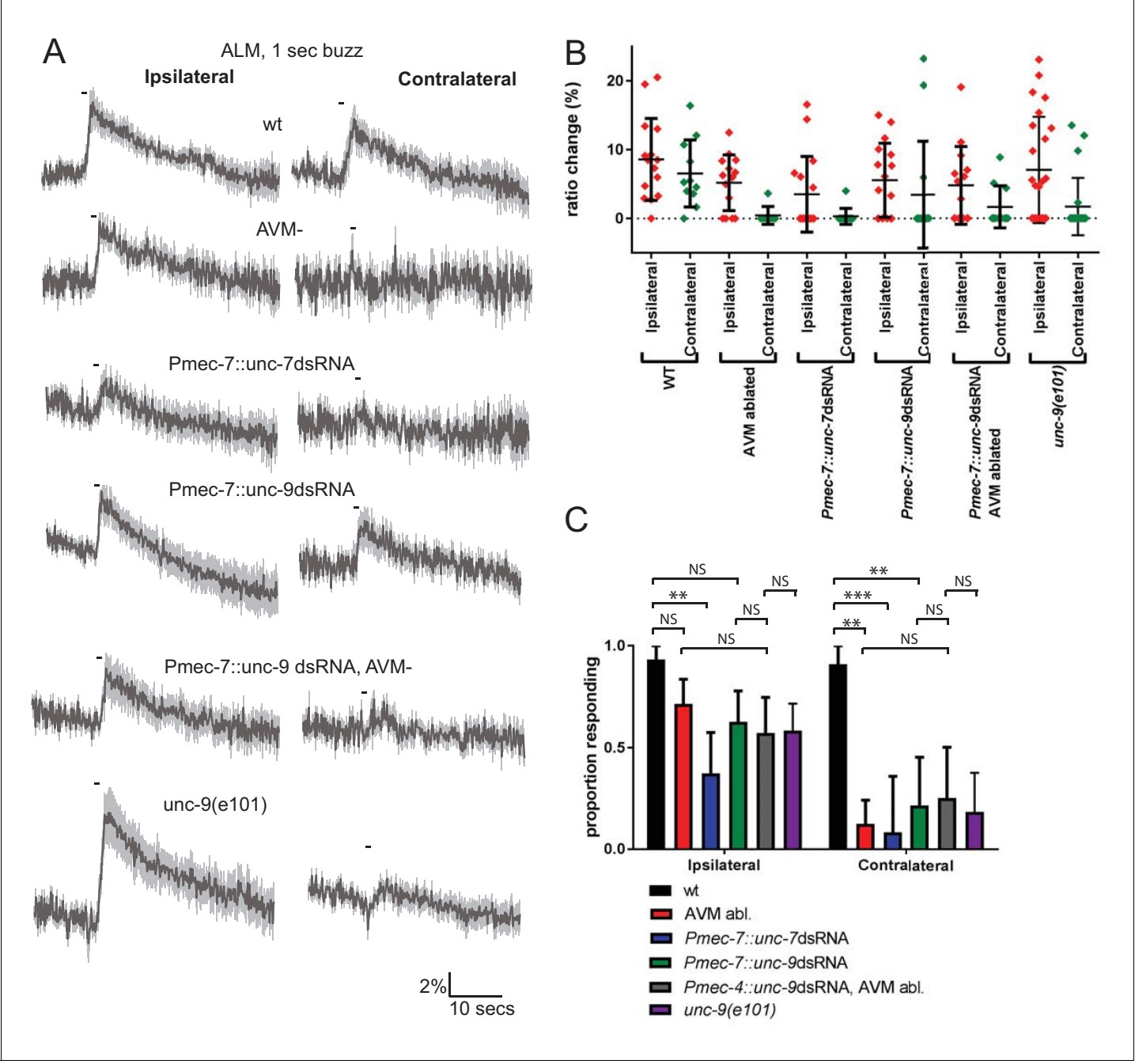

**Figure 2.** Innexins are required for mechanosensation and electrical coupling of touch neurons. (A, B, C) Gentle touch responses recorded in ALM for wild type worms, worms in which AVM has been laser ablated and worms expressing dsRNA under control of *Pmec-7*. Neurons have been classified as 'ipsilateral' or 'contralateral', according to the position of the cell body relative to the stimulation site and the hypothetical midline of the animal. (A) Average traces of % ratio change. Gray indicates SEM. (B) Scatter plot showing individual ratio changes (diamonds). Bars indicate mean ± SEM. (C) Graph showing proportion exhibiting a $Ca^{2+}$ response. Error bars indicate SE. In ipsilateral neurons, the proportion of AVM ablated animals (p=0.1686) or *unc-9* RNAi animals (p=0.1686) responding is not significantly different to wild type, while *unc-7* RNAi animals respond at a significantly reduced rate (p=0.0021). Combining *unc-9* RNAi with AVM ablation is not significantly different from either *unc-9* RNAi alone (p=0.7104), AVM ablation alone (p=0.6946) or *unc-9(e101)* (p=1). In contralateral neurons, the proportions of AVM ablated, *unc-7* RNAi and *unc-9* RNAi animals responding are all significantly lower than wild type (p=0.0012; p=0.0001; p=0.001). Combining *unc-9* RNAi with AVM ablation is not significantly different from either *unc-9* RNAi alone (p=1.0), AVM ablation alone (p=0.6027) or *unc-9(e101)* (p=0.661), Fisher's exact test (N = 15, 14, 16, 16, 14, 24, 11, 8, 12, 14, 11, 22, in the order shown in C).

The online version of this article includes the following figure supplement(s) for figure 2:

*Figure 2 continued on next page*

Figure 2 continued

**Figure supplement 1.** *unc-7* is required for gentle touch response in AVM.
**Figure supplement 2.** PLML and PLMR do not cooperate via gap junctions.

that despite the relatively low sequence similarity, UNC-7 and Pannexin one share significant functional conservation. Intriguingly, expression of a cDNA encoding a shorter *unc-7* isoform ('c' or UNC-7SR *Starich et al., 2009*) which is known to form gap junctions failed to rescue (*Figure 3A–C*). Thus, UNC-7's mechanosensory function appears to be genetically-separable from its ability to form gap junctions, and may therefore specifically involve hemichannels.

## *unc-7* is specifically required for mechanosensation in touch neurons and nociceptors

In principle, UNC-7 could affect mechanosensory responses by affecting the excitability of the touch neurons; alternatively, UNC-7 could play a direct role in mechanosensation. To address these possibilities, we examined the effect of *unc-7* knockdown and overexpression on channelrhodopsin-mediated activation of the TRNs. When channelrhodopsin is expressed in the TRNs, photostimulation evokes an escape response similar to those evoked by mechanosensory stimulation (*Nagel et al., 2005*). To assess whether *unc-7* RNAi affects TRN excitability, we chose a stimulus duration at which only two thirds of wild type animals responded. As expected, a *mec-4* null mutation did not significantly alter the proportion of animals responding, consistent with the specific role played by *mec-4* in mechanotransduction. Likewise, neither *unc-7* RNAi nor overexpression of 'cysless' *unc-7* significantly altered the proportion of animals responding to light stimulation (*Figure 4A*) suggesting that *unc-7* also does not alter the excitability of the touch neurons. The basal calcium activity of the touch neurons, as indicated by the baseline YFP/CFP ratio, also showed no significant difference between wild type animals (2.01 ± 0.22) and *unc-7(e5)* (1.82 ± 0.32). Together, these results indicate that loss of UNC-7 does not affect touch neuron excitability, and its role is likely to be specific to mechanosensation. *unc-7* is expressed in other sensory neurons, including the polymodal nociceptor PVD. PVD neurons respond to several aversive stimuli, including harsh touch and cold temperature (*Chatzigeorgiou et al., 2010*). To examine whether *unc-7* functions specifically in mechanosensation, we assayed the effect of *unc-7* mutations on both thermal and mechanical responses in PVD. We observed (*Figure 4B–D*) that *unc-7(e5)* animals were severely defective in the $Ca^{2+}$ response of the PVD neurons to harsh touch. In contrast (*Figure 4E–G*), *unc-7(e5)* animals showed no significant difference compared to wild-type in the PVD response to cold (temperature shift from 22° to 15°). Thus, *unc-7* is required for mechanosensory responses, but dispensable for thermosensory responses, in PVD, suggesting a specific role for UNC-7 in mechanotransduction.

The TRNs also exhibit responses to fast, high-displacement stimuli ('harsh touch') that are distinct from those seen in response to low-displacement press or buzz stimuli ('gentle touch'). Harsh touch responses are *mec-4*-independent, and are often (though not always) slower and longer-lasting than the responses observed for gentle touch stimuli (*Suzuki et al., 2003*; *Figure 5A*). Indeed, in response to a harsh stimulus, the responses of wild type animals can be sorted, based on the shape of the calcium trace, into 'transient' (similar to those seen for gentle touch, where the rise does not extend for more than 2 s beyond the stimulus period; these tend to be lower in amplitude) and 'prolonged' (harsh-specific responses, where the rise continues for an extended time beyond the stimulus period; these tend to be very high amplitude responses)(see *Figure 5A,B,C* and *Figure 5—figure supplement 1*). As observed previously (*Suzuki et al., 2003*) *mec-4* mutant animals showed a similar frequency of prolonged responses to wild-type (*Figure 5B,C* and *Figure 5—figure supplement 2*), although the frequency of transient responses was greatly reduced. In contrast, we found that *unc-7* RNAi specifically eliminated the prolonged responses (*Figure 5B,C* and *Figure 5—figure supplement 3*), while *unc-7(e5)* mutations eliminated virtually all TRN responses to harsh touch. Expression of cysless *unc-7* in the TRNs significantly rescued the harsh touch response defect in *unc-7* mutant animals, indicating that the loss of both transient and prolonged responses in the mutant was at least partially due to the cell-autonomous mechanosensory activity of UNC-7 hemichannels. *unc-7* knockdown in combination with *mec-4* null abolished both types of response, consistent with

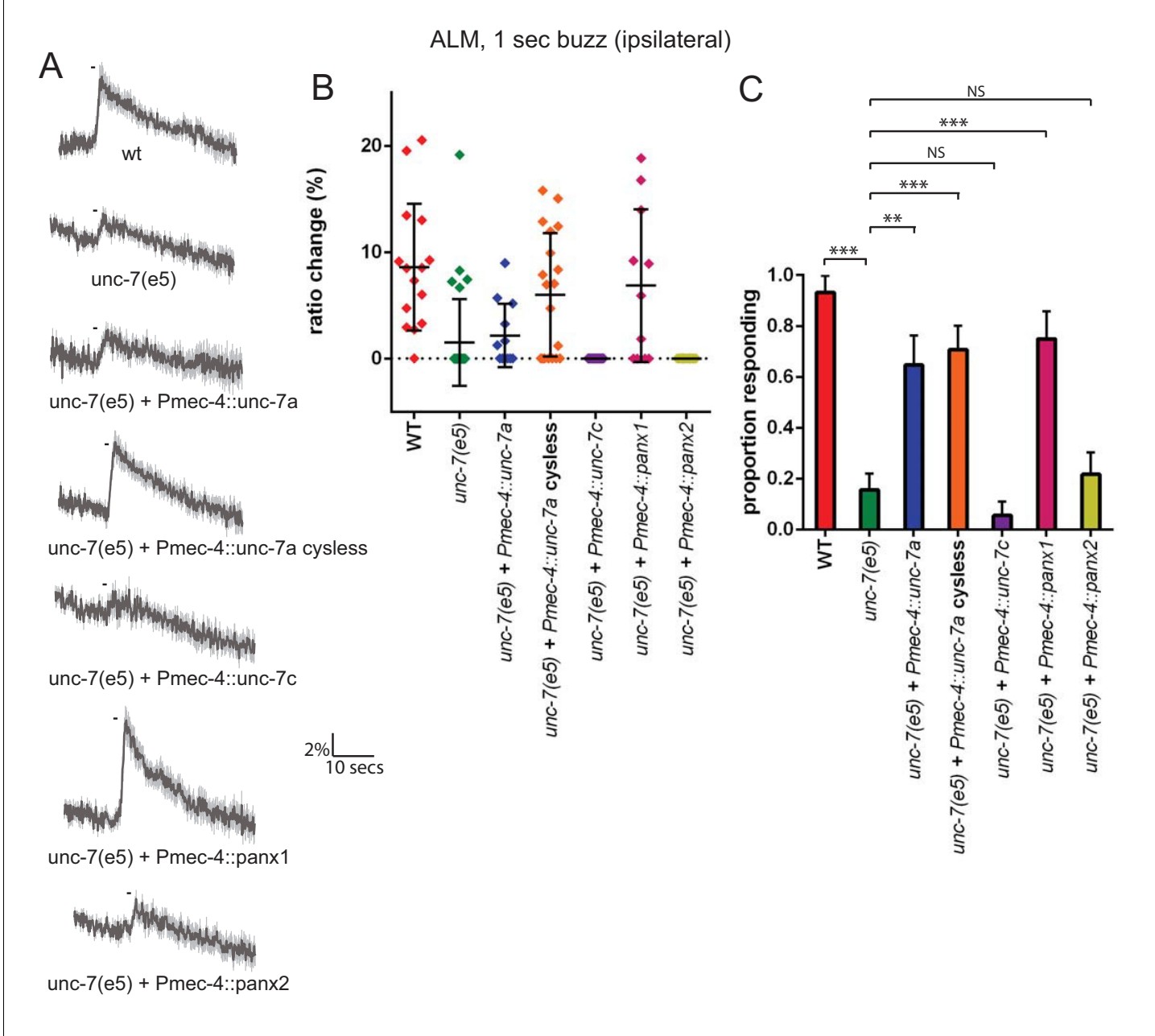

**Figure 3.** The mechanosensory function of UNC-7 is gap junction-independent. (A, B, C) Gentle touch responses recorded in ALM for wild type, *unc-7 (e5)*, and *unc-7(e5)* animals expressing *unc-7* isoforms or pannexins under the control of *Pmec-4*. 'Cysless' indicates C173A, C191A, C377A, C394A. All neurons recorded were ipsilateral, according to the position of the cell body relative to the stimulation site and the hypothetical midline of the animal. (A) Average traces of % ratio change. Gray indicates SEM. (B) Scatter plot showing individual ratio changes (diamonds). Bars indicate mean ± SEM. (C) Graph showing proportion exhibiting a $Ca^{2+}$ response. Error bars indicate SE. The response frequency is significantly reduced in *unc-7(e5)* compared to wildtype (p<0.0001). This is significantly rescued by TRN expression of wild type (p=0.001) or cysless *unc-7a* (p<0.0001). Cysless *unc-7* still significantly rescued the mutant when AVM was ablated (p<0.0001), and there was no significant difference between AVM ablated and unablated cysless *unc-7-*expressing animals (p=0.4701). While *unc-7c* (p=0.3991) and mouse *panx2* (p=0.7257) did not significantly rescue, *panx1* did (p<0.0001), Fisher's exact test (N = 15, 32, 12, 19, 13, 11, 18, in the order shown in the graphs).

The online version of this article includes the following figure supplement(s) for figure 3:

**Figure supplement 1.** Expression of *unc-7* or *unc-9* dsRNA in other neurons does not disrupt the gentle touch response.

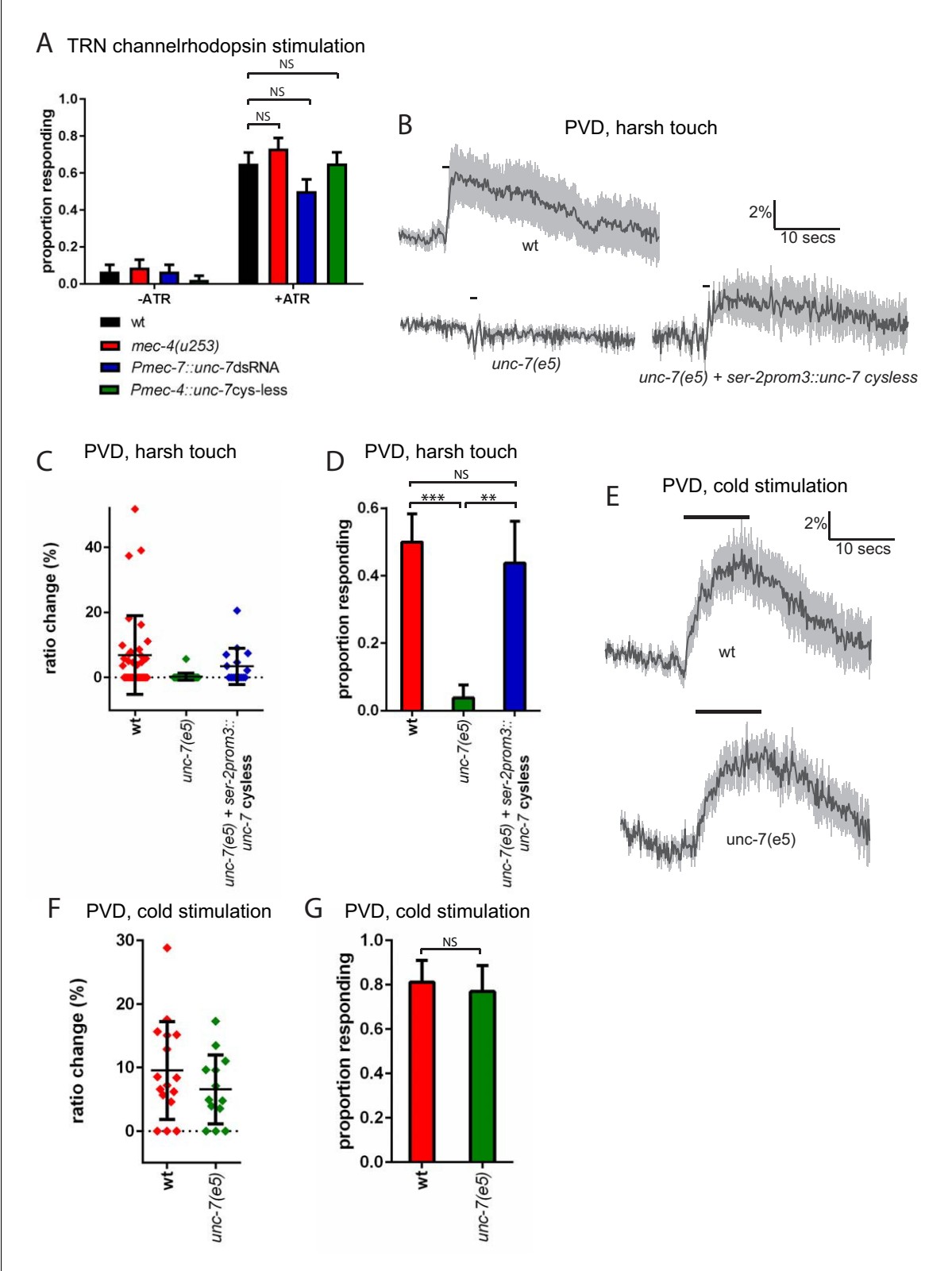

**Figure 4.** *unc-7* is specifically required for mechanosensation. (**A**) Behavioural response to light stimulation of animals expressing channelrhodopsin in the TRNs. The proportion of wild type animals responding was not significantly different to that for *Pmec-7::unc-7*dsRNA (p=0.1393), *Pmec-4::unc-7* cysless (p=1) or *mec-4(u253)* (p=0.4294) animals (N = 45, 45, 45, 45, 60, 60, 60, 60). All these experiments were carried out in a *lite-1* mutant background to eliminate effects of endogenous blue light responses (see strain list, **Supplementary file 1**). (**B,C,D**) Harsh touch responses recorded in PVD for wild

*Figure 4 continued on next page*

*Figure 4 continued*

type and *unc-7(e5)* animals. (**B**) Average traces of % ratio change. Gray indicates SEM. (**C**) Scatter plot showing individual ratio changes (diamonds). Bars indicate mean ± SEM. (**D**) Graph showing proportion exhibiting a $Ca^{2+}$ response. Error bars indicate SE. The proportion responding is significantly reduced in *unc-7(e5)* animals (p<0.0001); and this is significantly rescued (p=0.0026) to a response rate not significantly different (p=0.7683) from wild type (N = 36, 26, 16). (**E,F,G**) Cold responses recorded in PVD for wild type and *unc-7(e5)* animals. (**E**) Average traces of % ratio change. Black bar indicates shift from 22°C to 15°C. Gray indicates SEM. (**F**) Scatter plot showing individual ratio changes (diamonds). Bars indicate mean ± SEM. (**G**) Graph showing proportion exhibiting a $Ca^{2+}$ response. Error bars indicate SE. The proportion responding is not significantly different (p=1), Fisher's exact test, N = 16, 13).

the hypothesis that UNC-7 and MEC-4 act at least partially independently in the touch neurons, with UNC-7 but not MEC-4 particularly important for prolonged responses to harsh touch.

Although MEC-4 is required in the TRNs for neuronal responses to gentle touch (see for example *Figure 5—figure supplement 4*), the behavioural response to gentle touch is not completely lost in *mec-4(u253)* animals (*Nekimken et al., 2017*). To investigate whether UNC-7, perhaps acting locally in the ALM process, could account for this residual response, we assayed touch avoidance behaviour in *mec-4(u253)* animals expressing *unc-7* dsRNA in the TRNs. As expected, we observed (*Figure 5—figure supplement 5A*) that approximately a quarter of *mec-4* mutant animals responded behaviourally to gentle touch. However the combination of *mec-4(u253)* and *unc-7* RNAi did not eliminate this response; indeed, the phenotype resembled *mec-4(u253)* alone. Likewise, while disruption of *mec-4* or *unc-7* alone results in a very substantial decrease in the magnitude of the gentle touch response (i.e. the distance reversed; *Figure 5—figure supplement 5B*), *unc-7* RNAi did not further enhance this phenotype in a *mec-4* mutant background. Thus, additional neurons may be responsible for *mec-4*-independent touch avoidance; in the absence of *mec-4*, these might be enhanced by cross-modal plasticity (*Rabinowitch et al., 2016*).

## UNC-7 and MEC-4 act independently in touch neuron mechanosensation

To investigate the relationship between UNC-7 and MEC-4 in the touch neurons, we used fluorescently tagged transgenes to compare their intracellular localization patterns. As described previously (*Zhang et al., 2004*), mCherry-tagged MEC-4 protein was distributed in a punctate pattern along the ALM and PLM dendrites (*Figure 6A*). GFP-tagged UNC-7 was also expressed in a punctate pattern in both touch receptor neuron types. However, little overlap was observed between UNC-7 and MEC-4 puncta in either cell type (*Figure 6A,B*). Thus, UNC-7 does not appear to physically associate with MEC-4-containing mechanotransduction complexes, consistent with a distinct functional role in touch sensation.

Like *unc-7*, *mec-4* is critical for the response to gentle touch, and null mutations in *mec-4* result in an almost complete loss of touch-evoked $Ca^{2+}$ response (*Suzuki et al., 2003*). We reasoned that if UNC-7 hemichannels act independently of MEC-4, then their overexpression in the TRNs might compensate for the absence of MEC-4. Indeed, when cysless *unc-7* was overexpressed in the TRNs (*Figure 6C*) we observed strong suppression of the *mec-4(u253)* defect in the behavioural response to gentle touch. This suppression was independent of *mec-10*, the other DEG/ENaC known to function in the TRNs. Interestingly, *unc-7* overexpression only restored touch responses to stimuli applied near the head (a3, *Figure 1A*), and we were unable to detect any rescue of ALM $Ca^{2+}$ responses the *mec-4* mutant regardless of the site of stimulation (data not shown). This suggests that UNC-7 overexpression only partially compensates for loss of MEC-4, leading only to local depolarization of the ALM axon. Such activation might be insufficient to generate calcium transients in the ALM cell body, and only sufficient to activate downstream neurons and thus induce a behavioural response when triggered in presynaptic regions near the nerve ring. Conversely, we also tested whether overexpression of MEC-4 could compensate for the absence of UNC-7. We observed (*Figure 6D,E,F,G*) that when *mec-4* was overexpressed in the TRNs it strongly suppressed the behavioural and calcium defects in *unc-7(e5)* in response to either anterior or midbody touch stimuli. Thus, although both *mec-4* and *unc-7* are essential for the response to gentle touch in the TRNs, both can, at least to some extent, substitute for the other when overexpressed. This suggests that MEC-4 and UNC-7 indeed act independently as mechanotransducers in the TRNs.

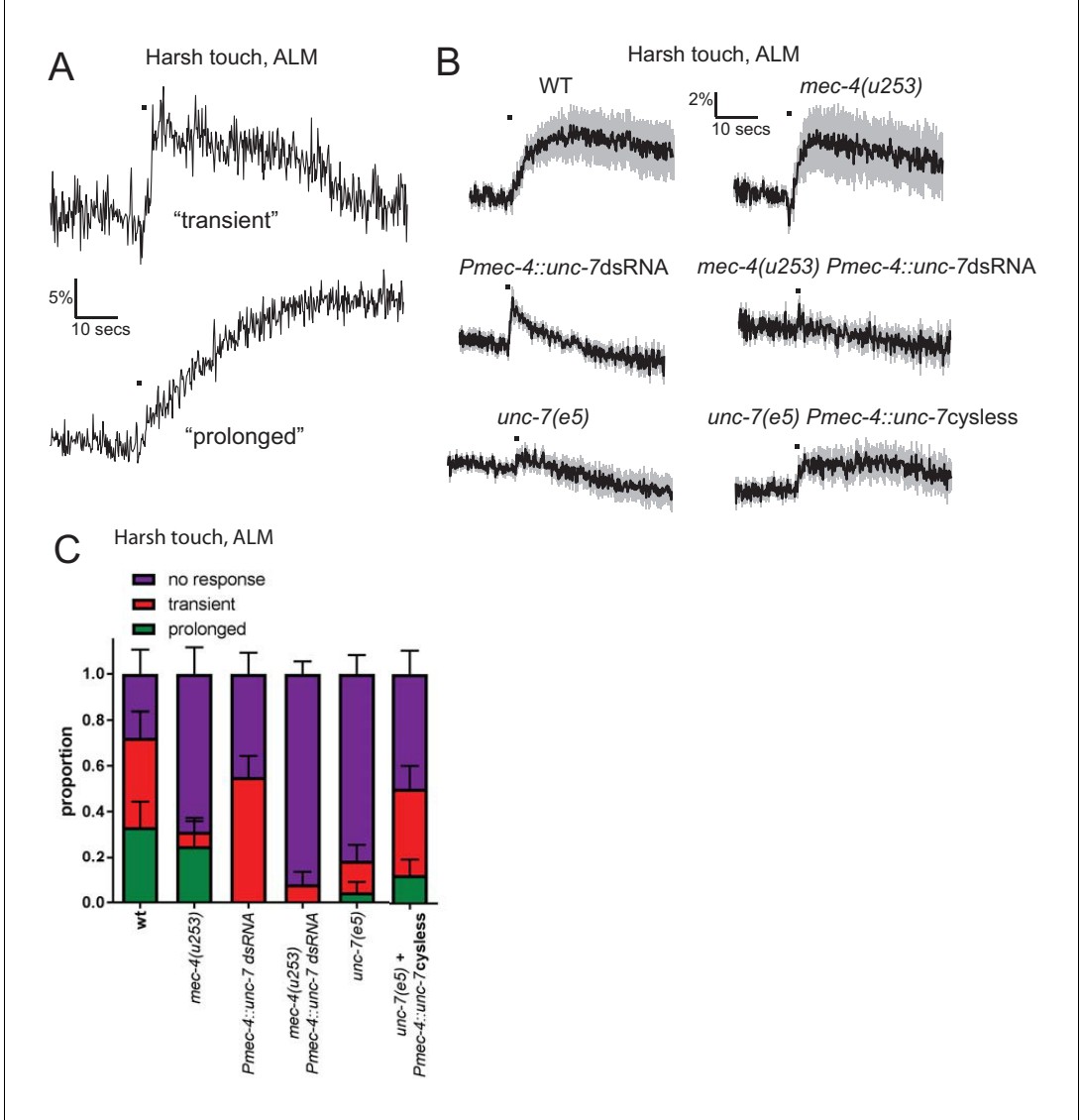

**Figure 5.** UNC-7 and MEC-4 have distinct roles in harsh touch. (**A, B, C**) Ca$^{2+}$ responses recorded in ALM in response to harsh touch. (**A**) Representative examples of the two types of Ca$^{2+}$ responses to harsh touch stimulation in ALM. (**B**) Average traces of % ratio change. Gray indicates SEM. (**C**) Proportion of animals displaying the indicated types of Ca$^{2+}$ response to harsh touch in ALM. Error bars are SE. The transient responses (characterized by a rapid onset and a decay to baseline beginning immediately after the stimulus ends) resemble typical gentle touch responses in the TRNs; prolonged responses (characterized by a slow onset that continues for several seconds following the end of the stimulus) is only seen in response to harsh touch. The transient responses are significantly disrupted in the absence of *mec-4* (p=0.0425), while the prolonged responses are significantly disrupted by *unc-7* knockdown (p=0.0328). *unc-7* RNAi, *mec-4* null combined completely abolishes both types of response (p=0.4898 when compared to zero responses; p=0.0001 when total response rate is compared to *unc-7* RNAi; p=0.0891 when compared to mec-4 null). *unc-7(e5)* significantly disrupts the total response rate (p=0.0011) and this is significantly rescued by TRN expression of *unc-7 cysless* (p=0.0324), Fisher's exact test (N = 18, 16, 29, 25, 22, 24 in the order shown on graphs).

The online version of this article includes the following figure supplement(s) for figure 5:

**Figure supplement 1.** Individual traces for the data shown in *Figure 5*, for wild type animals.

**Figure supplement 2.** Individual traces for the data shown in *Figure 5*, for *mec-4(u253)* animals.

**Figure supplement 3.** Individual traces for the data shown in *Figure 5*, *Pmec-7::unc-7* dsRNA animals.

**Figure supplement 4.** *mec-4* mutation and *unc-7* RNAi both severely disrupt the ALM response to gentle touch.

**Figure supplement 5.** *unc-7* RNAi disrupts the behavioural response to gentle touch.

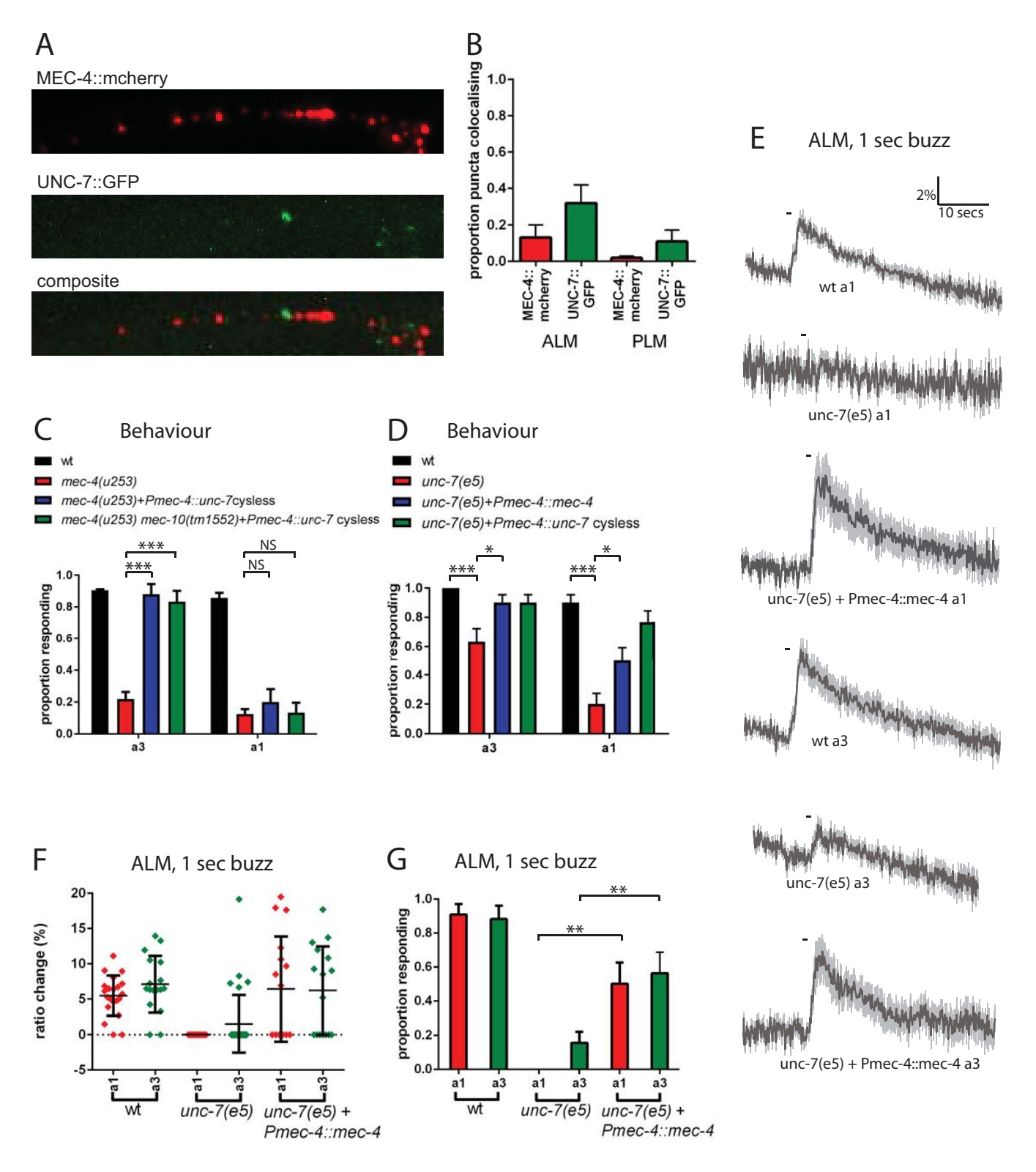

**Figure 6.** UNC-7 and MEC-4 act independently in touch neuron mechanosensation. (A, B) Confocal microscopy of TRN neurons expressing *mec-4*::mcherry and *unc-7a*::gfp. (A) Example images of PLM, and composite of the two channels, showing colocalisation in white. (B) Percentage of particles colocalising with particles of the other colour, based on centres of mass coincidence (N = 11, 11, 8, 8; total number of puncta = 161, 141, 156, 92). (C, D) Behavioural response to anterior gentle touch, for genotypes indicated. Animals were stimulated either at the back of the terminal bulb (a3) or

*Figure 6 continued on next page*

*Figure 6 continued*

approximately 50 μm anterior of the cell body of ALM (**a1**). Error bars are SE. TRN expression of *unc-7* cysless significantly rescued the behavioural defect of *mec-4(u253)*, including when *mec-10* was also defective, when stimulated at a3 (p<0.0001 for both); but not when stimulated at a1 (p=0.3423; p=1.0) (N = 40 for each genotype). *unc-7(e5)* animals are significantly defective in the behavioural response to gentle touch at a3 (p=0.0003) and a1 (p<0.0001), and TRN expression of *mec-4* significantly rescued this, at a3 (p=0.0303) and a1 (p=0.0292) (N = 30 for each genotype). (**E,F,G**) Ca$^{2+}$ responses to gentle touch recorded in ALM, for wild type, *unc-7(e5)*, and *unc-7(e5)* animals expressing P*mec-4::mec-4*. (**E**) Average traces of % ratio change. Light gray indicates SEM. (**F**) Scatter plot showing individual ratio changes (diamonds). Bars indicate mean ± SEM. (**G**) Graph showing proportion exhibiting a Ca$^{2+}$ response. Error bars indicate SE. Expression of *mec-4* significantly rescued the Ca$^{2+}$ response defect of *unc-7(e5)*, whether stimulated at a3 (p=0.0064) or a1 (p=0.0033), Fisher's exact test (N = 21, 17, 13, 32, 16, 16 in the order shown on graphs).

## Heterologous expression of UNC-7 hemichannels in olfactory neurons confers touch sensitivity

Our results so far indicate that UNC-7 is necessary for normal mechanosensation in the TRNs. If UNC-7 hemichannnels play a direct role in mechanotransduction, they might also be expected to be sufficient to confer mechanosensory responses in cells that are natively touch-insensitive. To test this possibility, we expressed the cysless derivative of *unc-7* in the ASKs or the ASJs, two pairs of ciliated amphid neurons that do not respond to mechanical stimulation (*Figure 7*). Although at least one study has reported native *unc-7* expression in ASK, there is no evidence for such expression in ASJ (*Bhattacharya et al., 2019*). We then assayed mechanosensory activity potentially conferred by the heterologously expressed transgene by measuring touch-evoked neural activity using an ASK-expressed genetically-encoded calcium indicator.

When we expressed the cysless *unc-7* transgene alone, we observed robust nose touch responses in ASK that were absent in the ASK neurons of wild-type animals (*Figure 7A,B,C*). In contrast, expression of *mec-4* in the same way did not render ASK mechanically sensitive, even when coexpressed with *mec-2*. Coexpression of either *mec-2* or *mec-4* with cysless *unc-7* did not enhance the ectopic touch responses in ASK; indeed, the responses of coexpressing animals were if anything smaller than those of animals expressing *unc-7* alone. When we expressed the cysless *unc-7* transgene in ASJ, we observed small but significant nose touch responses that were absent in the ASJ neurons of wild-type animals (*Figure 7D,E,F*). These results are consistent with the hypothesis that UNC-7 hemichannels are sufficient to form a mechanosensor, though we cannot rule out the possibility that heterologous UNC-7 expression enhances an endogenous touch-sensitive response that is undetectable in wild-type neurons. In either case, UNC-7 may require few if any additional specific factors to carry out its mechanosensory function, whereas MEC-4 appears to require additional proteins to generate a mechanotransduction complex.

## Discussion

### UNC-7 hemichannels function specifically in mechanosensation

We have shown here that the innexin UNC-7 plays an essential role in the response to gentle touch, and that this mechanosensory function is likely mediated by hemichannels rather than gap junctions. Several lines of evidence support these conclusions. First, loss of *unc-7* function affects touch responses to ipsilateral stimuli as well as to contralateral stimuli, implying its role is not merely in indirect activation through gap junctions. Second, *unc-7* mutations lead to mechanosensory defects in neurons such as the PLMs, which are not known to be connected by gap junctions, and this may also be the case for the PVDs. Third, an *unc-7* transgene containing mutations that render it incapable of gap junction formation still effectively rescues the *unc-7* touch-insensitive phenotype. Our observation that *unc-7* functions affects mechanosensory, but not thermosensory responses in the PVD polymodal nociceptors, coupled with the fact that *unc-7* does not impair channelrhodopsin-mediated excitation of the TRNs, provide evidence that the UNC-7 hemichannels specifically affect mechanosensory transduction rather than general neuronal excitability. Heterologous expression of UNC-7 hemichannels in two other neuron classes, ASK and ASJ also suggested a specific role in mechanotransduction.

How might UNC-7 contribute to mechanotransduction in the touch neurons and PVD nociceptors? Perhaps the simplest hypothesis is that UNC-7 hemichannels are themselves mechanically-

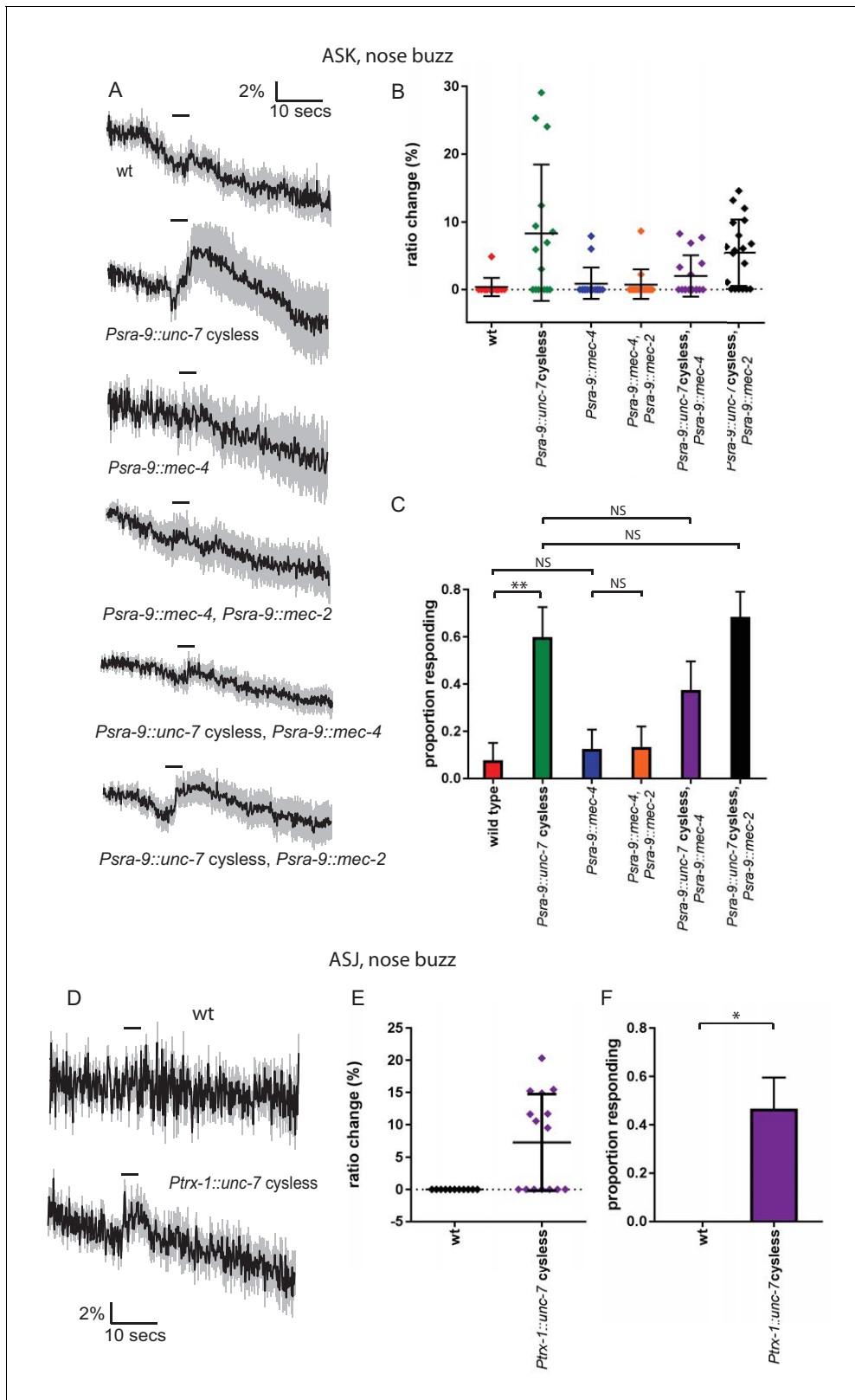

**Figure 7.** Heterologous expression of UNC-7 hemichannels in olfactory neurons confers touch sensitivity. (A, B, C) Nose touch responses recorded in ASK of wild type animals and animals expressing *unc-7* cysless or *mec-4* in ASK. (A) Average traces of % ratio change. Light gray indicates SEM. (B) Scatter plot showing individual ratio changes. Bars indicate mean ± SEM. (C) Graph showing proportion exhibiting a Ca$^{2+}$ response. Error bars indicate SE. Wild type ASK neurons do not significantly respond to nose touch (p=1.0), but expression of *unc-7* cysless significantly increases the response rate

*Figure 7 continued on next page*

*Figure 7 continued*

(p=0.006). Expression of *mec-4* does not significantly increase the response rate (p=1.0), and coexpression of *mec-4* does not significantly alter the response rate for *unc-7* cysless expressing animals (p=0.2890). Coexpression of *mec-2* does not significantly increase the response rate for *mec-4* or *unc-7*, Fisher's exact test (N = 13, 15, 16, 15, 16, 19 in the order shown on graphs). (D, E, F) Nose touch responses recorded in ASJ of wild type animals and animals expressing *unc-7* cysless in ASJ. (D) Average traces of % ratio change. Light gray indicates SEM. (E) Scatter plot showing individual ratio changes. Bars indicate mean ± SEM. (F) Graph showing proportion exhibiting a $Ca^{2+}$ response. Error bars indicate SE. Wild type ASJ neurons do not significantly respond to nose touch (p=1), but expression of *unc-7* cysless significantly increases the response rate (p=0.0103), Fisher's exact test (N = 11, 15).

gated ion channels whose opening contributes to the mechanoreceptor potential. Consistent with this possibility, pannexin 1, a mammalian homologue of UNC-7 that functionally complements its touch phenotype in worms, has been shown to form mechanosensitive channels when expressed in *Xenopus* oocytes (*Bao et al., 2004*). However, although heterologously-expressed UNC-7 (in the cysless mutant form) appears to have channel activity, the potential mechanosensitivity of these channels was not reported (*Bouhours et al., 2011*). Alternatively, UNC-7 hemichannels might play an accessory role in mechanosensation; for example, they might amplify the mechanoreceptor potential, or modulate the primary mechanotransducer by mediating transient changes in calcium or other messengers (*Vanden Abeele et al., 2006*). In the future, physiological characterization of UNC-7 hemichannel properties may distinguish between these hypotheses.

Although these results are the first to implicate UNC-7 as a mechanosensory molecule, other results are consistent with a role for innexins in *C. elegans* touch sensing. For example, it was shown recently (*Sangaletti et al., 2014*) that the TRNs express a mechanically gated current with innexin-like physiological and pharmacological properties. However, the mechanically gated currents that they identified are intact in *unc-7(e5)* animals, indicating that UNC-7 is not an essential component of this particular current (R. Sangaletti and L. Bianchi, personal communication). It is unclear how the pneumatic pressure stimulus used in these studies relates to externally-applied gentle touch, and one possibility is that different mechanically sensitive innexins function over different sensitivity ranges.

## UNC-7 and MEC-4 function independently in touch neurons

Unexpectedly, we have found that two ion channels, UNC-7 and MEC-4, are both required for normal touch responses in the TRNs; loss-of-function of either UNC-7 or MEC-4 alone leads to significant touch insensitivity. Nonetheless, several lines of evidence suggest that UNC-7 and MEC-4 function independently in the touch neurons, rather than functioning together in a common mechanotansduction complex. First, although both UNC-7 and MEC-4 proteins are distributed in a punctate pattern along the TRN dendrite, UNC-7- and MEC-4-containing puncta do not colocalize, and therefore appear to represent physically-distinct complexes. Second, although *unc-7* and *mec-4* single mutants both are insensitive to gentle touch, overexpression of *mec-4* can suppress the *unc-7* defect, while *unc-7* overexpression can partially suppress the gentle touch defect of *mec-4*. Third, *mec-4* and *unc-7* mutations have distinct phenotypes with respect to harsh touch responses in the TRNs; *unc-7* affects and is required for large, long-lasting *mec-4* independent responses, whereas *mec-4* is required for small, transient responses that resemble responses to gentle touch. Thus, although the activities of both UNC-7 and MEC-4 appear to be essential for sensitivity to weaker stimuli, they function at least somewhat redundantly in the response to stronger stimuli such as harsh touch.

What does this imply about the function of UNC-7 in mechanosensation? One possibility is that UNC-7 and MEC-4 are each mechanically-sensitive channels that respond to qualitatively distinct types of mechanical stimulation. MEC-4, for example, is believed to be tethered to both the extracellular matrix and the cytoskeleton (*Arnadóttir and Chalfie, 2010*), whereas UNC-7, like the bacterial Msc, might directly sense membrane tension via lipid interactions, (*Kung et al., 2010*). These different force detection mechanisms might in principle confer distinct biomechanical properties, resulting in specificity in the precise mechanical forces to which they respond. If neuronal response to gentle stimuli requires coincident activation of both UNC-7 and MEC-4, due to summing of these distinct inputs, this might also serve to filter out noise and improve the fidelity of response to small but behaviourally significant stimuli. In this context, it is interesting to note the recent demonstration

(*Servin-Vences et al., 2017*) that both TRPV4 and PIEZO are required for mechanosensation in chondrocytes, and that they appear to function in distinct ways: PIEZO responds to membrane stretch, while TRPV4 appears to rely on tensile forces transmitted via the matrix.

Alternatively, UNC-7 might not itself be a mechanosensitive channel, but rather might modulate touch neuron responses by amplifying mechanoreceptor potentials or locally enhancing excitability. This accessory function might be essential to enhance MEC-4-dependent gentle touch responses, but partially dispensable, at least in the presence of MEC-4, for responses to stronger harsh touch stimuli. According to this model, UNC-7 overexpression might suppress the *mec-4* gentle touch phenotype by sensitizing or enhancing the *mec-4*-independent harsh touch mechanotransducer in the TRNs. Interestingly, UNC-7 appears to be necessary for the slow and prolonged time course typically observed in harsh touch responses; thus, perhaps this property results from prolonged opening of UNC-7 channels following the stimulus.

## Coordination of the anterior TRNs via gap junctions

In addition to its role in mechanosensation, UNC-7, along with UNC-9, also contributes to gap junctions that functionally link the anterior TRNs. In the locomotion circuit, for example in the electrical synapses between the AVB premotor interneurons and the B-class motorneurons, UNC-7 and UNC-9-containing gap junctions appear to be asymmetric, with UNC-7 expressed in AVB and UNC-9 expressed in B motor neurons (*Starich et al., 2009*). Likewise, UNC-7 and UNC-9 also form heterotypic gap junctions between AVA and A motor neurons (*Kawano et al., 2011*). In contrast, in the touch neurons, both innexins have been reported to be expressed in both AVM and ALM (*Altun et al., 2009*; *Starich et al., 2009*), and the *unc-7* mechanosensory phenotypes likewise imply expression in both ALM and AVM. Thus, for the ALM-AVM gap junctions, it seems likely that UNC-7 and UNC-9 contribute in both partner cells. Unlike UNC-7, the role of UNC-9 appears to be confined to gap junctions, since *unc-9* mutations (like AVM ablations) only affect responses to contralateral stimuli. It is interesting that an analogous cooperative organisation (i.e. connection of PLML and PLMR via PVM) does not exist for the posterior TRNs, and our hypothesis that this results in a functionally distinct network is corroborated by our demonstration that PLM responses are significantly less robust when the neurons are far from the stimulation site. The difference in organisation presumably reflects the dominance of forward locomotion, and thus the greater selective pressure on the avoidance of anterior aversive stimuli.

An important feature of any sensorimotor circuit must be to ensure that a given stimulus elicits an appropriate behavioural response. Encountering an innocuous obstacle requires a distinct response (brushing against it or a steering change) compared to a noxious stimulus (a rapid avoidance movement). We have seen previously in the nose touch circuit that electrical connections between sensory neurons can play an important role; when OLQ and CEP are activated by gentle nose touch, they facilitate gentle touch responses in FLP, which otherwise only responds to harsh touch, whereas if OLQ and CEP are inactive, they inhibit the activity of FLP through shunting (*Chatzigeorgiou and Schafer, 2011*; *Rabinowitch et al., 2013*). Since the FLPs provide the link to the premotor interneurons, the result is that only a harsh stimulus or a broad gentle stimulus generates an escape response, whereas more localised gentle stimuli generate distinct behaviours (head withdrawal; food slowing), depending on which of these neurons are stimulated. The case of the anterior gentle touch neurons is different, in that all three (ALML, ALMR, AVM) synapse directly onto premotor neurons. Nevertheless, an attractive hypothesis is that a similar electrically-coupled circuit amplifies the behavioural response when all anterior touch neurons are coincidently activated.

## UNC-7 plays genetically-distinct roles in gap junctions and mechanosensory hemichannels

Although UNC-7 appears to function in both gap junctions and mechanosensory hemichannels, these two functions are genetically separable. For example, we have shown that whereas the L isoform of UNC-7 can rescue the mechanosensory defect of *unc-7(e5)*, the shorter SR isoform fails to rescue. Conversely, others (*Starich et al., 2009*) have shown that UNC-7SR and S can restore gap junction activity in the locomotion circuit, whereas UNC-7L could not rescue gap junction-dependent behavioural phenotypes and that UNC-7L could not produce gap junction currents in vitro. Since UNC-7L has been shown to form hemichannels when expressed in neuro2A cells (*Bouhours et al.,*

*2011*), this suggests a distinction between isoforms, with UNC-7SR and S being required for gap junctions, while UNC-7L is capable of forming hemichannels that participate in mechanosensation. The three isoforms differ only in the length of the N-terminal cytoplasmic region (see *Starich et al., 2009* for full details). The N-terminal region that is unique to UNC-7L is rich in proline, indicating a potential role in interaction with WW domain-containing proteins (*Kay et al., 2000*), and several other amino acid repeats that are suggestive of protein interaction motifs. Thus, this extended domain might mediate differential localisation or trafficking of UNC-7, or alternatively could interfere, either directly or through inter-protein interactions, with assembly into gap junctions. It is interesting to note that UNC-7L is the only isoform expressed in the ALM, AVM and PVD neurons (*Starich et al., 2009*), while reporters specific for the shorter forms or lacking UNC-7L-specific upstream elements lack some or all mechanosensory neuron expression (*Altun et al., 2009*; *Bhattacharya et al., 2019*). Likewise, recent evidence (*Bhattacharya et al., 2019*) indicates that UNC-7 is expressed in ASK, and yet this neuron is mechanically insensitive, and can be rendered mechanically sensitive by expression of UNC-7. A logical explanation for this apparent paradox would be that the isoform natively expressed in ASK is the shorter UNC-7SR (or UNC-7S). If UNC-7L is indeed the only isoform that plays a role in mechanosensation, this unique domain could also hold the key to understanding the molecular basis of this function.

## Functional conservation of UNC-7 with mammalian pannexins

Intriguingly, we observed that the mechanosensory function of *unc-7* could be complemented by a mammalian homologue, the *Panx1* pannexin gene. Expression of a mouse *Panx1* transgene fully rescued the touch-insensitive defect of an *unc-7* null mutant, indicating strong functional conservation across a large phylogenetic distance. Although pannexins have not been directly implicated in touch or other somatosensory processes in vertebrates, Panx1 has been shown to be mechanically sensitive, mediating the release of ATP in a variety of cell types in response to membrane stretch. Our finding of a role for UNC-7, and its functional complementation by Panx1, suggests the possibility that pannexins might play undiscovered roles in touch or other mechanical senses in vertebrates.

Pannexins have been implicated in a huge array of medical conditions, including ischaemia-induced seizure, inflammation, hypertension, tumour formation and metastasis, and neuropathic pain, and are thus an important target for therapeutic intervention. A significant obstacle is their involvement in so many functions, requiring a deep understanding of these roles in order to intervene specifically. However, even their basic properties (non-selective versus anion-selective; high conductance versus low conductance) remain controversial (*Chiu et al., 2014*; *Good et al., 2015*). Our observation that mouse pannexin one can be functionally expressed in *C. elegans* neurons opens the door to a tractable model organism in which to study pannexin itself. As UNC-7 fulfils multiple functions in different cell types, understanding how this is determined will provide clues as to how this is achieved for pannexins in higher organisms.

# Materials and methods

## *C. elegans* strains

Strains used in this study are described in *Supplementary file 1*.

*Plasmid constructs.* P*mec-7*::dsRNA plasmids for *unc-7*, *unc-9*, *inx-7* were constructed by ligating a cDNA fragment of approximately 600 bp between the second and third multiple cloning sites of pPD117.01 (A gift from Andrew Fire). The complementary sequences used in the primers were as follows: *unc-7*: gttgctacgtcactatgctc and agtctatcgtcccttgaccg; *unc-9*: atgctattgtattatttcgcg and agtcgttgagaacttgcagtc; *inx-7*: tcgtgtcttaaacactgttcc and agaatcttgtgtggaactatc. An *E. coli Cat1* (chloramphenicol acetyl transferase gene) dsRNA plasmid was constructed in a similar fashion. For each target, two plasmids, with sense and antisense orientations of the insert, were co-injected at 50 ng/μl each. P*nmr-1*::dsRNA plasmids were constructed by replacing P*mec-7* in these plasmids with P*nmr-1* (1.7 kb). *unc-7* rescue plasmids were made using the Multisite Gateway 3-Fragment Vector Construction Kit (Invitrogen). A 1078 bp *mec-4* promoter fragment (as previously used, *Suzuki et al., 2003*), was cloned into pDONR P4-P1R. *unc-7* (isoforms a and c) cDNA were amplified from RB1 cDNA library (a gift from Robert Barstead) and cloned into pDONR 221. These were combined with pDONR P2R-P3/SL2::mcherry (a gift from Mario de Bono) in a derivative of pDEST R4-R3 into which

unc-54 3'UTR had been inserted downstream of the recombination sites. Cysteine to arginine substitutions were made in the relevant pDONR 221 plasmid, using codon-optimised mutagenic primers, designed using *C. elegans* Codon Adapter (http://worm-srv3.mpi-cbg.de/codons/cgi-bin/optimize.py; *Redemann et al., 2011*). Partially overlapping complementary mutagenic primers were used to amplify the plasmid using Phusion High-Fidelity DNA Polymerase (Thermo Scientific), then *Dpn*I digestion was used to remove bacterially-derived template DNA, before transformation into *E. coli*. These were assembled into the *Pmec-4* expression vector in the same way as the wild type sequences. Pannexin 1 and 2 were amplified from a mouse cDNA library, and assembled into the *Pmec-4* expression vector in the same way. PVD and ASK expression vectors were constructed using the same Gateway strategy, using *ser-2prom3* and *sra-9* promoters, respectively (gifts from Marios Chatzigeorgiou and Lorenz Fenk; *Chatzigeorgiou et al., 2010*; *Fenk and de Bono, 2015*). The plasmids were injected at 50 ng/µl. TRN-specific *mec-4*::mcherry and *unc-7a*::gfp fusions were constructed in a similar way, using the *mec-4* promoter, and injected at 20 ng/µl and 50 ng/µl, respectively. A second *unc-7a*::gfp encoded a fusion, where GFP was inserted in the internal loop, between N290 and I291, surrounded by Gly Gly linkers, exhibited the same localisation.

## Ca²⁺ imaging

$Ca^{2+}$ imaging of anterior and posterior body touch stimulation of glued animals was essentially as described previously (*Kerr et al., 2000*; *Suzuki et al., 2003*), using a 1 s 'buzz' stimulus just posterior of the terminal bulb. For gentle touch, the probe displacement was 10 µm; for harsh touch it was delivered using a glass needle with a sharper end, a displacement of 30 µm and a faster velocity, 2.8 µm/s. To avoid habituation, care was taken that animals were not repeatedly stimulated. They were allowed to acclimatise for at least 5 min following mounting and where an animal was stimulated more than once, the animal was allowed to recover for at least 2 min. Posterior harsh body touch (for PVD stimulation) and nose 'buzz' stimulation (for ASK stimulation) were performed as described previously (*Chatzigeorgiou et al., 2010*; *Kindt et al., 2007*). Images were recorded at 10 Hz using an iXon EM camera (Andor Technology), captured using IQ1.9 software (Andor Technology) and analysed using Spikefinder and Neurontracker (see Supplemental materials), Matlab (MathWorks) analysis scripts, written by Ithai Rabinowitch (*Rabinowitch et al., 2013*). Illumination levels were below that required to evoke a blue light calcium response. For thermosensation, animals were glued in the usual way then treated by perfusion of buffer at the temperatures indicated. Mechanical stimulation for TRN and PVD imaging was performed in Neuronal Buffer (145 mM NaCl, 5 mM KCl, 5 mM CaCL₂, 5 mM MgCl₂, 20 mM glucose, 10 mM HEPES, pH7.2). Thermal stimulation and nose touch stimulation were performed in CTX (25 mM KPO₄ pH6, 1 mM CaCl₂, 1 mM MgSO₄). Where appropriate, neurons were categorised into ipsilateral or contralateral, depending on the position of their cell body with respect to the site of stimulation and a hypothetical midline.

## Channelrhodopsin experiments

L4 animals were transferred to retinal plates (made by seeding 55 mm NGM plates with 160 µl of a 1000:4 mixture of OP50 culture and 100 mM all-*trans* retinal in ethanol and incubating overnight at 22°C), and grown at 22°C, then assayed as day one adults. Assay plates were prepared by seeding 30 mm NGM plates with 40 µl of a 1000:1 mixture of OP50 culture and 100 mM all-*trans* retinal in ethanol and incubating for 30 min at 22°C. Control plates contained ethanol without all-*trans* retinal. Animals were picked individually to assay plates using an eyelash hair, and acclimatised for 15 min. To stimulate the TRNs, *lite-1(ce319)* worms expressing *Pmec-4::ChR2* (*Rabinowitch et al., 2016*) were illuminated for 1 s with 1 mW/mm² blue light using 470 nm LEDS controlled by a LEGO Mindstorms Intelligent NXT Brick.

## Laser killing of AVM

Laser ablation was carried out in L1/L2 animals as described by *Bargmann and Avery (1995)*.

## Behavioural assays

Gentle and harsh body touch were performed on day one adults, by stroking with an eyelash hair or prodding with a platinum wire pick, respectively (*Chalfie, 2014*) just behind the pharynx terminal

bulb (a3, *Figure 1A*), except where stated. For *Figure 5—figure supplement 5*, the length of reversal was quantified by counting head swings (*Li et al., 2011*).

## Confocal microscopy

Images were acquired using a Zeiss LSM 780. Colocalisation was visualised using the Colocalization Finder plugin (C Laummonerie and J Mutterer, Strasbourg, France) for ImageJ. Object-based colocalisation of puncta was analysed using the JACoP plugin (*Bolte and Cordelières, 2006*) for ImageJ, using particle centre of mass coincidence. The functionality of fusion protein transgenes was verified by checking their ability to rescue null mutants.

## Acknowledgements

We are very grateful to Yee Lian Chew, Kristin Webster and Yiquan Tang for critical reading of the manuscript, and members of the Schafer, de Bono and Taylor labs for helpful discussions. We are grateful to the LMB workshops for help with the channelrhodopsin setup. We thank Andrew Fire, Robert Barstead, Lorenz Fenk, Mario de Bono and Marios Chatzigeorgiou for plasmids and strains. Some strains were provided by the CGC, which is funded by NIH Office of Research Infrastructure Programs (P40 OD010440). Neurontracker and Spikefinder were written by Ithai Rabinowitch. This work was funded by the Medical Research Council (MC_A023_5PB91) and Wellcome Trust (WT103784MA).

## Additional information

### Funding

| Funder | Grant reference number | Author |
| --- | --- | --- |
| Medical Research Council | MC-A023-5PB91 | William R Schafer |
| Wellcome | WT103784MA | William R Schafer |
| National Institutes of Health | 1R21DC015652 | William R Schafer |

The funders had no role in study design, data collection and interpretation, or the decision to submit the work for publication.

### Author contributions

Denise S Walker, Conceptualization, Data curation, Formal analysis, Validation, Investigation, Visualization, Methodology; William R Schafer, Conceptualization, Supervision, Funding acquisition

### Author ORCIDs

Denise S Walker [ID] https://orcid.org/0000-0003-1534-1679
William R Schafer [ID] https://orcid.org/0000-0002-6676-8034

### Decision letter and Author response

Decision letter https://doi.org/10.7554/eLife.50597.sa1
Author response https://doi.org/10.7554/eLife.50597.sa2

## Additional files

### Supplementary files

- Source code 1. SpikeFinder 4.4.
- Source code 2. NeuronTracker 3.1.
- Supplementary file 1. Strains used in this study.
- Transparent reporting form

## Data availability

All data generated or analysed during this study are included in the manuscript.

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
