## [Decision Letter]

**Acceptance summary:**

Mechanosensation mediating touch- pain- or proprioceptive sense is essential for animals. The nematode *C. elegans* dedicates a remarkable proportion of its nerve cells (at least ~10%) to perceiving such mechanical stimuli. Worms exhibit conserved molecular machineries for mechanosensation, like channel complexes of the DEG/ENaC and TRP families. Vertebrates also employ hemichannels of the pannexin family for mechanosensation; their invertebrate relatives, the innexins, however, were thought to act primarily as gap junctions. Here, Walker and Schafer show that the major innexins in *C. elegans*, termed UNC-7 and UNC-9, might serve indeed such dual roles. They report evidence suggesting that UNC-9 is involved in electrical coupling in a network of a few touch receptor neurons (TRNs), while UNC-7 also exhibits the role as a mechanosensitive hemichannel in TRNs. Strikingly, the *unc-7* gene can be functionally substituted with the vertebrate pannexin gene *Panx1*, suggesting a common mechanism for mechanosensation shared between innexins and pannexins.

**Decision letter after peer review:**

Thank you for submitting your article "Distinct roles for innexin gap junctions and hemichannels in mechanosensation" for consideration by *eLife*. Your article has been reviewed by three peer reviewers, one of whom is a member of our Board of Reviewing Editors, and the evaluation has been overseen by Ronald Calabrese as the Senior Editor. The following individual involved in review of your submission has agreed to reveal their identity: Mei Zhen (Reviewer #2).

The reviewers have discussed the reviews with one another and the Reviewing Editor has drafted this decision to help you prepare a revised submission.

Summary:

In this manuscript Walker and Shafer study a potential role of the innexin *unc-7* gene in *C. elegans* mechanosensation. Based on their results they propose that *unc-7* might serve as a hemichannel directly involved in primary sensory transduction, either alone or in complex with other yet un-identified components. In their model, innexin genes serve two roles in anterior touch receptive neurons (TRNs). (1) integration of bilaterally received stimuli via a gap junction network of ALMR-AVM-ALML, and (2) direct mechanosensation via *unc-7* hemichannels. They collect support for this model via tissue specific dsRNAi and transgenic rescue experiments. Their model finds the strongest support in their finding that a mutant *unc-7* variant, previously shown incapable of electrical coupling ("cysless *unc-7*"), can rescue some of the mechanosensory defects of *unc-7* null mutants. Moreover, *unc-7* is not required for TRN excitability via ChR2 activation suggesting specificity. Ectopic expression of *unc-7* renders ASK neurons mechanosensitive, suggesting sufficiency in the ASK context. Interestingly, *unc-7* function can be rescued with a mammalian *pannexin1* gene (there is some evidence in the literature that *pannexin1* might function as a stretch sensitive hemichannel). Direct electrophysiological and biophysical evidence that UNC-7 is indeed a hemichannel or mechanosensitive is missing in this study, therefore in the evidence is indirect. However, in total many lines of indirect come together to support the major claims. The results are surprising, considering that *unc-7* expression encompasses a large portion of the worm's nervous system and that it was typically seen as a major gap junction gene. The work has impact on the interpretation of previous studies, assuming a sole role for *unc-7* in electrical coupling. Moreover, it paves way for interesting future work studying how *unc-7* exactly confers mechanosensitivity.

Essential revisions:

1) Reviewer 1. The authors rely on previous gene expression studies (e.g. Althun, 2009), however, these data have been outdated by a recent state of the art expression study using fosmid reporters (Bhattacharya, 2019). Here, *unc-7* seems not to be expressed in AVM and ALM in adult animals. This is confusing, and the authors ignore this recent study, not even citing it. Since the authors' conclusions strongly depend on endogenous *unc-7* expression in anterior TRNs and since this seems controversial now it is necessary to validate expression of *unc-7* in AVM and ALM in adult animals using proper reporter constructs, showing that they overlap with a well characterized TRN marker.

2) Reviewer 1. *unc-7* and *unc-9* are expressed in many neurons of the worm and ALM and AVM form gap junctions with other neurons, therefore non-cell autonomous effects of their gene knock-downs need to be controlled for. The authors apply a cell specific dsRNAi technique. dsRNAi, however, potentially could spread from cell-to-cell (e.g. Jose et al., 2009, PNAS). Since the author's interpretations strongly depend on the assumption of TRN-specific knockdown, additional controls are needed. The most straight forward control would be to rescue the RNAi phenotype with a codon altered RNAi insensitive transgene in TRNs.

2b) Reviewer 2. The key evidence here was that RNAi against UNC-9 in TRN showed modest or no change in the calcium response of ALM and PLM, in contrast to the drastic effect of RNAi against UNC-7. My concern is the lack of information or control experiments that demonstrate the specificity of UNC-7 RNAi, and the robustness of UNC-9 RNAi. Both were driven by the same promoter. There is sequence similarity between the two cDNAs. It was unclear which 600bp sequences were selected for each innexin to address their specificity and efficiency. One useful control would be to compare the effect of both RNAi driven pan-neuronally, on how effective they are to mimic the phenotypes of *unc-7* or *unc-9* mutants.

3) Reviewer 1. I find the AVM requirement for contralateral ALM responses quite interesting; this important result goes a bit under in the Discussion. However, in the remaining experiments shown in Figure 2, the authors claim a differential effect in *unc-9* and *unc-7* mutants. They argue that *unc-9* mutation has a specific effect on contralateral responses while *unc-7* affects both, ipsilateral as well as contralateral responses. They conclude right away that *unc-9* is mainly involved in gap junction coupling and *unc-7* additionally in primary mechano-transduction. Looking at the traces in 2A and scatter graphs in 2B, these statements are somewhat on weak grounds. All manipulations in 2B seem to follow the same trend, especially when examining the scatter graphs. The authors rely solely in their p-values obtained, but to be more convincing simply more repetitions are needed. In any case, it is possible that *unc-7* and *unc-9* function in a very similar manner, simply dsRNAi could be differentially effective. Without controlling that both are complete knockdowns, the authors should be more careful with concluding differential roles for *unc-7* vs. *unc-9*.

4) Reviewer 2. I found some results here hard to interpret. In *unc-7* mutants, all touch response was abolished, including the response to harsh touch by all TRNs and PVD. MEC-4 specifically affects gentle touch and is the proposed gentle touch sensor. It is difficult to envision why overexpression of MEC-4 could robustly bypass the requirement of UNC-7, especially in the harsh touch response (and I am not sure if authors addressed this directly). A simpler interpretation of these results is that the expression of cation leak channels (at ER or plasma membranes) changes a neuron's intrinsic property. If authors have attempted to address if MEC-4 can specifically rescue the harsh touch-mediated response in PVD and ALM in the *unc-7* mutants. MEC-4 is not responsible for harsh-touch mediated (behavioral response at least). If MEC-4 overexpression still rescues, then it is more likely that the cation leak non-specifically changed PVD's sensitivity to mechanical stimulation.

5) Reviewer 2. Authors should address the impact of a recent paper (Bhattacharya et al., 2019), which suggest that both UNC-7 and UNC-9 are endogenously expressed in ASK, on the interpretation of this experiment. Calcium-imaging results from this experiment appeared highly variable (Figure 7A, B). An overexpression of UNC-7 may alter ASK's property, and its network property (e.g. altering the strength of GJ coupling with other neurons in the case of both overexpressing wild-type or Cysless UNC-7). The authors may want to select another neuron that does not have UNC-7 expression to test their hypothesis again.

6) Reviewer 3. Please discuss a potential for controversy with previously published data: one of the major questions that remained unanswered, and to my surprise aren't even discussed, is the following: If *unc-7* acts independently of *mec-4*, why isn't there any evidence of mechanoreceptor current in the absence of *mec-4* in electrophysciological recordings (e.g. Ohagan et al). Also, no calcium transients were observed in *mec-4* null mutants in other works (Nekimken et al., 2017), even with Ca indicators superior to the one used in this present study. A potential explanation would be that *unc-7* is only able to generate local transients and thus remain undetectable for stimuli far away from the cell body, however, in previous studies neither were current or transients recorded after stimuli close to the cell body.

7) Reviewer 3. *mec-7p* is not specific to touch receptor neuron but also expressed in head neurons and probably PVD. Thus, the statement of cell-specificity needs to be toned down and concerns arise that the observed effects are due to defect in inx of other neurons. The authors addressed this in parts by using a *mec-4p* construct already.

8) Reviewer 3. How often has the same animal been tested? Are there still calcium signals in animals lacking *mec-4* under the presented experimental conditions for gentle touch? How do the authors distinguish between these two modalities – any quantitation would help to reconcile their conclusion.

9) Reviewer 3. In previously published data, deletion of *mec-4* still causes a response to gentle touch (e.g. Vasquez, Cell Reports, 2014 or Nekimken, 2017) but seems to habituate after the first or second touch. This is puzzling as no mechanoreceptor currents could be measured using electrophysiology or visualized in dendrites with calcium reporter. The authors also report behavioral response in the *mec-4* mutation, consistent with a function of another channel in this response – did they try the *mec-4/unc-7* double mutation to see if the animals did not respond at all?

---

## [Author Response]

Essential revisions:1) Reviewer 1. The authors rely on previous gene expression studies (e.g. Althun, 2009), however, these data have been outdated by a recent state of the art expression study using fosmid reporters (Bhattacharya, 2019). Here, unc-7 seems not to be expressed in AVM and ALM in adult animals. This is confusing, and the authors ignore this recent study, not even citing it. Since the authors' conclusions strongly depend on endogenous unc-7 expression in anterior TRNs and since this seems controversial now it is necessary to validate expression of unc-7 in AVM and ALM in adult animals using proper reporter constructs, showing that they overlap with a well characterized TRN marker.

We apologize for the confusion here; in fact only *unc-9* was shown to be expressed in the anterior touch neurons in the Altun et al., 2009 paper (which used short promoter fusions). The expression of *unc-7* in anterior touch neurons (specifically the long form) was reported in Starich et al., which used a cosmid-based reporter which (based on physical maps; the 5’ breakpoint of neither construct has been precisely determined) contained even more upstream sequence than the fosmid-based construct used by Bhattacharya. The expression pattern was specifically shown in that paper to mirror the staining pattern of an anti-*unc-7* antibody. Bhattacharya also did not report expression in some other neurons that expressed long form-specific reporters in the Starich manuscript. Thus, we suspect that the Bhattacharya construct lacks some upstream elements necessary for expression of the long form of *unc-7*. Single cell transcriptional profiling (Cao et al., 2017) also provides evidence that *unc-7* is expressed in the TRNs. We note that the fact that touch neuron-specific RNAi experiments (but not off-target controls) phenocopy the *unc-7* loss-of-function mutant, which also implies that *unc-7* is endogenously expressed in touch neurons. It is interesting that according to the Starich paper PVD also expresses only the long-form of *unc-7*, which we show to be the only isoform of *unc-7* rescues mechanosensory phenotypes.

We have fixed the citations in the Introduction (fourth paragraph) and added a discussion of the expression patterns of various isoforms and reporters (including the results of Bhattacharya) and their relation to mechanosensory activity (subsection “UNC-7 plays genetically-distinct roles in gap junctions and mechanosensory hemichannels”).

2) Reviewer 1. unc-7 and unc-9 are expressed in many neurons of the worm and ALM and AVM form gap junctions with other neurons, therefore non-cell autonomous effects of their gene knock-downs need to be controlled for. The authors apply a cell specific dsRNAi technique. dsRNAi, however, potentially could spread from cell-to-cell (e.g. Jose et al., 2009, PNAS). Since the author's interpretations strongly depend on the assumption of TRN-specific knockdown, additional controls are needed. The most straight forward control would be to rescue the RNAi phenotype with a codon altered RNAi insensitive transgene in TRNs.

We have added off-target RNAi controls (Figure 3—figure supplement 1) showing that expression of double-stranded *unc-7* RNA in other neuron types (premotor neurons) does not lead to touch phenotypes. This is consistent with our expectations since the SID complex required for intracellular uptake of small RNAs (Jose et al., 2009) is not expressed in neurons (Winston et al., 2002), and other investigators have found RNAi to act cell-specifically within the nervous system (Esposito et al., 2007 and many papers citing this).

We note also that the cell-specificity of the mechanosensory phenotypes in ALM (Figure 3) and PVD (Figure 4) were also shown by cell-specific rescue of the *unc-7(e5)* loss-of-function mutant. We have expanded the text (subsection “Innexins are required for mechanosensation and electrical coupling of touch neurons”) to make it clearer that we have used *Pmec-7* for RNAi and the more specific *Pmec-4* for the rescue experiments. We have also edited the description of the initial RNAi results to avoid giving the impression that we are claiming this is evidence of a cell-autonomous role in the TRNs (subsection “The mechanosensory function of UNC-7 is gap junction-independent”).

2b) Reviewer 2. The key evidence here was that RNAi against UNC-9 in TRN showed modest or no change in the calcium response of ALM and PLM, in contrast to the drastic effect of RNAi against UNC-7. My concern is the lack of information or control experiments that demonstrate the specificity of UNC-7 RNAi, and the robustness of UNC-9 RNAi. Both were driven by the same promoter. There is sequence similarity between the two cDNAs. It was unclear which 600bp sequences were selected for each innexin to address their specificity and efficiency. One useful control would be to compare the effect of both RNAi driven pan-neuronally, on how effective they are to mimic the phenotypes of unc-7 or unc-9 mutants.

To address this question, we have included imaging experiments on the *unc-9* loss-of-function mutant; as with the *unc-9* RNAi line, these animals showed no abnormality in touch response per se, but a defect coupling between the ALM lateral pair (revised Figure 2). We have included the primer sequences in the Materials and methods (subsection “Plasmid constructs”). Contiguous sequence conservation between *unc-7* and *unc-9* is not substantial; for the *unc-7* construct, the longest contiguous stretch of conserved bases is 9, for *unc-9* it is 17.

3) Reviewer 1. I find the AVM requirement for contralateral ALM responses quite interesting; this important result goes a bit under in the Discussion. However, in the remaining experiments shown in Figure 2, the authors claim a differential effect in unc-9 and unc-7 mutants. They argue that unc-9 mutation has a specific effect on contralateral responses while unc-7 affects both, ipsilateral as well as contralateral responses. They conclude right away that unc-9 is mainly involved in gap junction coupling and unc-7 additionally in primary mechano-transduction. Looking at the traces in 2A and scatter graphs in 2B, these statements are somewhat on weak grounds. All manipulations in 2B seem to follow the same trend, especially when examining the scatter graphs. The authors rely solely in their p-values obtained, but to be more convincing simply more repetitions are needed. In any case, it is possible that unc-7 and unc-9 function in a very similar manner, simply dsRNAi could be differentially effective. Without controlling that both are complete knockdowns, the authors should be more careful with concluding differential roles for unc-7 vs. unc-9.

The possibility that differential RNAi efficacy could explain the differences between ipsilateral and contralateral responses has been addressed by comparing the phenotypes of the *unc-7* and unc-9 null mutants. As outlined in the response to reviewer 2, comment 2b above, the *unc-9(e101)* mutant (revised Figure 2) shows the same phenotype as *unc-9* RNAi. We confined our statistical comparisons to the proportion responding because we were concerned that disruption of gap junctions could have confounding effects (i.e. a lack of shunting through gap junctions might result in an increase in amplitude) which could mask the low proportion of responders.

The role of *unc-9* in touch neuron gap junction connectivity has indeed been somewhat ignored in the Discussion! We have inserted an additional section (subsection “Coordination of the anterior TRNs via gap junctions”) and reorganised the section headings to address this.

4) Reviewer 2. I found some results here hard to interpret. In unc-7 mutants, all touch response was abolished, including the response to harsh touch by all TRNs and PVD. MEC-4 specifically affects gentle touch and is the proposed gentle touch sensor. It is difficult to envision why overexpression of MEC-4 could robustly bypass the requirement of UNC-7, especially in the harsh touch response (and I am not sure if authors addressed this directly). A simpler interpretation of these results is that the expression of cation leak channels (at ER or plasma membranes) changes a neuron's intrinsic property. If authors have attempted to address if MEC-4 can specifically rescue the harsh touch-mediated response in PVD and ALM in the unc-7 mutants. MEC-4 is not responsible for harsh-touch mediated (behavioral response at least). If MEC-4 overexpression still rescues, then it is more likely that the cation leak non-specifically changed PVD's sensitivity to mechanical stimulation.

This is an interesting point. We have tried the experiment suggested by the reviewer in the TRNs and found that *mec-4* overexpression does not rescue the harsh touch defect of the *unc-7* mutant. We have not included these data in the figure since the reciprocal experiment (overexpression of UNC-7 in a *mec-4* mutant) would not be meaningful given the robust harsh touch response of *mec-4* single mutants, but in total only 14.3% responded, all with a low amplitude “transient” response, N=14. We have not tried this experiment in PVD because neither MEC-4 nor many of its required cofactors (e.g. MEC-9) are natively expressed there and it would therefore be difficult to interpret the results.

We agree that the *mec-4* overexpression results do not argue conclusively that UNC-7 itself is a MEC-4-independent mechanotransduction channel. In the revision, we have modified the text in many places to clarify that UNC-7 hemichannels might play other roles in touch sensing, including as a channel that alters local excitability as suggested by the reviewer. We have particularly addressed this with regard to the overexpression experiments in the subsection “Coordination of the anterior TRNs via gap junctions”).

5) Reviewer 2. Authors should address the impact of a recent paper (Bhattacharya et al., 2019), which suggest that both UNC-7 and UNC-9 are endogenously expressed in ASK, on the interpretation of this experiment. Calcium-imaging results from this experiment appeared highly variable (Figure 7A, B). An overexpression of UNC-7 may alter ASK's property, and its network property (e.g. altering the strength of GJ coupling with other neurons in the case of both overexpressing wild-type or Cysless UNC-7). The authors may want to select another neuron that does not have UNC-7 expression to test their hypothesis again.

To address this point, we expressed UNC-7 in the ASJ neurons, one of the few chemosensory neurons that show no evidence of *unc-7* expression in the Bhattacharya paper. We did observe small but significant nose touch responses in the *unc-7*-expressing lines that were absent in control animals. These results have been added to revised Figure 7. Since these responses were small and therefore not inconsistent with other interpretations, we have added a caveat in the subsection “Heterologous expression of UNC-7 hemichannels in olfactory neurons confers touch sensitivity”. As we have shown that at least one of the shorter *unc-7* isoforms does not confer mechanosensitivity, expression of such an isoform in ASK may explain the native touch-insensitivity of ASK neurons (subsection “Functional conservation of UNC-7 with mammalian pannexins”).

6) Reviewer 3. Please discuss a potential for controversy with previously published data: one of the major questions that remained unanswered, and to my surprise aren't even discussed, is the following: If unc-7 acts independently of mec-4, why isn't there any evidence of mechanoreceptor current in the absence of mec-4 in electrophysciological recordings (e.g. Ohagan et al.). Also, no calcium transients were observed in mec-4 null mutants in other works (Nekimken, 2017), even with Ca indicators superior to the one used in this present study. A potential explanation would be that unc-7 is only able to generate local transients and thus remain undetectable for stimuli far away from the cell body, however, in previous studies neither were current or transients recorded after stimuli close to the cell body.

This is a good point, and conforms quite closely to our interpretation of the results in Figure 6. Specifically, while MEC-4 overexpression was able to compensate for the cell body calcium imaging as well as the behavioural phenotype of *unc-7* mutants, UNC-7 overexpression did not restore cell body calcium transients to *mec-4* null mutants and only rescued the behavioural phenotype for stimuli applied near the animal’s neck. As suggested by the reviewer, a simple explanation of this is that UNC-7 overexpression only allows local depolarization of the ALM dendrite and that MEC-4 is necessary for cell-body responses to gentle touch. We have modified the text to make this point more explicitly (subsection “UNC-7 and MEC-4 act independently in touch neuron mechanosensation”f).

We note that while *mec-4* null mutants appear to lack mechanosensory responses to gentle touch in ALM, both our lab (Suzuki et al., 2003, Chatzigeorgiou et al., 2010, this study Figure 5B) and Hang Lu’s lab (Cho et al., LoC 2017, Cho et al. LoC 2018) have observed robust responses to stronger mechanical stimuli in *mec-4* null animals (using either cameleon or GCaMP6, and with either glued or microfluidic immobilization/stimulation). (The chip used in the Nekimken paper gave a much lower response rate even in wild-type animals than what we observe; thus this is likely to represent a very gentle stimulus; the stimulus in the O’Hagan paper on dissected animals likewise may represent a gentler stimulus.) So if *unc-7* encodes either another mechanosensor or a specific amplifier of mechanotransduction currents, its overexpression could potentially enhance sensitivity of those harsh touch responses. We have added a comment to this effect to the text (subsection “UNC-7 and MEC-4 function independently in touch neurons”).

7) Reviewer 3. mec-7p is not specific to touch receptor neuron but also expressed in head neurons and probably PVD. Thus, the statement of cell-specificity needs to be toned down and concerns arise that the observed effects are due to defect in inx of other neurons. The authors addressed this in parts by using a mec-4p construct already.

We used the *mec-7* promoter in initial RNAi experiments because it is a very strong promoter that expresses in touch neurons, though as the reviewer notes it is less selective than *mec-4*. We have clarified this in the text (subsections “Innexins are required for mechanosensation and electrical coupling of touch neurons” and “The mechanosensory function of UNC-7 is gap junction-independent”). With regard to the cell autonomy of the *unc-7* mechanosensory phenotypes, as noted in the response to reviewer 1 comment 2 we have confirmed all the key experiments using an *unc-7* loss-of-function mutant rescued using the *mec-4* promoter.

8) Reviewer 3. How often has the same animal been tested? Are there still calcium signals in animals lacking mec-4 under the presented experimental conditions for gentle touch? How do the authors distinguish between these two modalities – any quantitation would help to reconcile their conclusion.

Care was taken that animals were not repeatedly stimulated; where they were stimulated more than once, a wait ensured that we avoided habituation effects. The protocols for gentle and harsh body touch are now explained in the Materials and methods section (previously we just cited the original references) (subsection “Plasmid constructs”).

We have included calcium imaging for gentle touch for strains used in Figure 5, in Figure 5—figure supplement 4. In agreement with previous reports, we observe only occasional calcium responses in *mec-4* null animals following gentle touch stimulation (subsection “*unc-7* is specifically required for mechanosensation in touch neurons and nociceptors”).

9) Reviewer 3. In previously published data, deletion of mec-4 still causes a response to gentle touch (e.g. Vasquez, Cell Reports, 2014 or Nekimken, 2017) but seems to habituate after the first or second touch. This is puzzling as no mechanoreceptor currents could be measured using electrophysiology or visualized in dendrites with calcium reporter. The authors also report behavioral response in the mec-4 mutation, consistent with a function of another channel in this response – did they try the mec-4/unc-7 double mutation to see if the animals did not respond at all?

This is a really interesting point. In common with these previous reports, we also see a residual behavioural response in the *mec-4* mutant (26% respond, although the length of the reversal is substantially diminished), We carried out the experiment suggested by the reviewer, assaying the gentle touch avoidance response in *mec-4* null mutant animals in which *unc-7* was knocked down by RNAi. We observed that these *mec-4/unc-7* doubly-deficient animals still responded to gentle touch at frequencies similar to the *mec-4*-single mutant, indicating that the MEC-4-independent behavioural response is not mediated by UNC-7; this suggests a role for another neuron class in this behaviour. We have presented behavioural data in Figure 5—figure supplement 5 and discussed this in the Results section (subsection “*unc-7* is specifically required for mechanosensation in touch neurons and nociceptors”).